# IAmHero: Preliminary Findings of an Experimental Study to Evaluate the Statistical Significance of an Intervention for ADHD Conducted through the Use of Serious Games in Virtual Reality

**DOI:** 10.3390/ijerph20043414

**Published:** 2023-02-15

**Authors:** Annamaria Schena, Raffaele Garotti, Dario D’Alise, Salvatore Giugliano, Miriam Polizzi, Virgilio Trabucco, Maria Pia Riccio, Carmela Bravaccio

**Affiliations:** 1Villa delle Ginestre s.r.l. Rehabilitation and FKT Centre, 80040 Volla, Italy; 2Unità Operativa Semplice Dipartimentale of Child Neuropsychiatry, Department of Translational Medical Sciences, 80131 Naples, Italy

**Keywords:** ADHD, executive function, IAmHero, serious game, virtual reality

## Abstract

The use of new technologies, such as virtual reality (VR), represents a promising strategy in the rehabilitation of subjects with attention-deficit/hyperactivity disorder (ADHD). We present the results obtained by administering the IAmHero tool through VR to a cohort of subjects with ADHD between 5 and 12 years of age. The trial time was approximately 6 months. In order to assess the beneficial effects of the treatment, standardised tests assessing both ADHD symptoms and executive functions (e.g., Conners-3 scales) were administered both before and at the end of the sessions. Improvements were observed at the end of treatment in both ADHD symptoms (especially in the hyperactivity/impulsivity domain) and executive functions. One of the strengths of the VR approach is related above all to the acceptability of this tool and its flexibility. Unfortunately, to date, there are still few studies on this topic; therefore, future studies are essential to expand our knowledge on the utility and benefits of these technologies in the rehabilitation field.

## 1. Introduction

Attention-deficit/hyperactivity disorder (ADHD) is a neurodevelopmental disorder characterised by persistent patterns of inattention and/or hyperactivity-impulsivity that interfere with functioning or development. This condition is characterized by marked levels of inattention; hyperactivity (excessive, persistent, and continuous motor activity); and difficulty in controlling behavioral and verbal impulses. Children with ADHD seem to be always engaged in some activity, although they often do not complete it as they are constantly distracted by new stimuli. The tendency not to listen and/or excessive motor activity result in restlessness, difficulty sitting, and inability to wait their turn. These manifestations (hyperactivity, impulsivity, and inattention) are nothing more than the consequence of the inability of the child with ADHD to control his or her responses to stimuli arising from the environment and to focus his or her attention on a single specific task [1]. The diagnosis of ADHD is established by meeting certain clinical criteria. Specifically, to define attention-deficit/hyperactivity disorder, symptoms must occur for 6 consecutive months in at least two different life contexts (such as, for example, school and family); in addition, these symptoms must present since childhood (specifically before the age of 12 years) [2]. The causes of ADHD are not yet fully known, but the origin of the disorder seems to depend on a combination of environmental, social, behavioural, biochemical, and genetic factors [1]. This disorder is often comorbid with other neurodevelopmental disorders, and its treatment involves different therapeutic approaches (cognitive behavioural therapy, psychotherapy, and psychopharmacology) [3]. ADHD is a neurodevelopmental disorder with a strong impact on the lives of affected individuals: there are difficulties both in social and academic skills, with impairment in quality of life in general [4,5]. Treatment can be both non-pharmacological [1] and pharmacological (with stimulating agents such as methylphenidate or non-stimulating agents such as atomoxetine) [6]. The literature shows that the multimodal approach combining both behavioural therapy and stimulant therapy is more effective than other combinations or single treatments [7]. Further difficulties in the therapeutic approach are related to the different presentation of symptoms in the course of growth (e.g., greater hyperactivity in childhood, the persistence of inattention in adolescence and adulthood) but also to the emergence of comorbidities (e.g., oppositional defiant disorder, conduct disorder, substance abuse disorder) [8]. In the last decade, information and communication technologies (ICT) have offered new application possibilities and new perspectives for intervention in this field. Possible behavioural approaches include the use of serious games: these are an emerging technology to improve attentional skills and collaborative behaviour. This new approach has often been suggested in the literature as an innovative method in the treatment of patients with ADHD [9,10,11]. Although children with ADHD are expected to present difficulties in engaging in recreational activities due to their poor attention span, they can sometimes focus for long periods of time on pleasurable activities, according to a phenomenon known as “hyperfocus” [12]. Therefore, a rehabilitative approach conducted in the form of video games would seem to improve the cooperation of such patients and enable a better course of treatment [13]. A serious game is defined as an interactive simulation having the structure of a video game but whose purpose is to develop and/or enhance skills, knowledge or attitudes that can be transferred to the real world. Serious games synergistically and effectively combine three main elements: (a) simulation, since they allow reproducing real events or some aspects of them; (b) learning, since the latter is the main and ultimate goal of the serious game itself; and (c) gaming, since the serious game, being in effect a video game, includes elements of gaming (e.g., storyline, interactivity, challenge, and explorable environment) that motivate and engage the player. Serious games facilitate several learning processes, including experiential learning (learning by doing). This is because the player can experience and train in specific contexts, but at the same time, he or she can accurately observe both the context, the various details, and the time sequence of actions to be performed. These observations activate the visual and motor memory mechanisms that promote the ability to generalize knowledge at later times and in different contexts. A guided learning experience is thus proposed to the player in which he or she can reflect on and metabolize the simulative game activity experienced (learning by reflexion) [14]. These are digital games that combine educational and playful elements. The proposed games aim to improve specific cognitive and behavioural skills [15], identified in collaboration with teams of medical experts and therapists. These games are realised in an immersive reality environment and involve the use of advanced technologies, such as virtual reality (VR) and motion-sensing equipment, which allow both to increase the engagement and flow state of the patient and to detect patient performance parameters [16,17,18]. The aim of our study is to evaluate the effectiveness of IAmHero, administered through VR methods, on a cohort of subjects with ADHD. There are different studies in the literature regarding the use of such methods not only in patients with ADHD but in general in the rehabilitation of psychiatric disorders [19,20,21]. One of the critical issues present, for example, is the paucity of studies that have evaluated these studies by inter-group comparisons (treated vs. untreated) [19]. To this end, we believe it is important to be able to suggest the use of an additional tool that may be useful in improving both the core symptoms of ADHD and the executive functions of such subjects. Despite the presence of some evidence in the literature, there are still few studies on this topic to be able to unequivocally define the use of VR in the field of rehabilitation.

## 2. Material and Methods

### 2.1. Study Sample

Sixty children belonging to the Villa delle Ginestre rehabilitation center in Volla (NA) and the U.O.S.D. NPI of the Department of Translational Medical Sciences, University of Naples Federico II took part in the study. The eligibility criteria used are shown in Table 1. 

In contrast, the exclusion criteria for the study were the presence of physical impairments and/or known organic syndromes and/or unstabilized pediatric neurocomorbidities (e.g., seizures) or medical status (e.g., diabetes mellitus).

Participants were randomly divided into two groups, one experimental and one control, and were given virtual reality therapies with the IAmHero system alternating with traditional therapy and traditional therapies only, respectively. There were 33 male and 27 female participants, and the average age was 8 years. Eighty percent of the participants had a diagnosis of ADHD without comorbidity, while of the remaining 20 percent, 18 percent had comorbidity with specific learning disorders.

### 2.2. Experimental Design and Clinical Assessment

The study included an initial phase in which participants were enrolled through a diagnostic screening at the U.O.S.D. of Child Neuropsychiatry of the Department of Translational Medical Sciences, University of Naples Federico II. In this phase, the following tests were administered: the Wechsler Intelligence Scale for Children 4th edition (WISC-IV) for the assessment of the cognitive profile; the Italian Battery for ADHD (BIA), aimed at the assessment of attention and concentration skills; the Conners-3 questionnaire for the assessment of ADHD and behavioural disorders; and the Tower of London test (TOL) to assess strategic decision-making and problem-solving skills [22]. These tests were used for diagnostic confirmation as well as a definition of pre-intervention cognitive and attentional abilities. This was followed by a pre-intervention phase, the purpose of which was to assess the participants’ performance with the help of specific tests and scales to define clinical outcomes. This phase also involved the U.O.S.D. of Child Neuropsychiatry of the Department of Translational Medical Sciences, University of Naples Federico II. The intervention subsequently involved 30 participants, with a chronological age range of 5–12 years, undergoing training with serious games at our unit and 30 participants undergoing conventional therapy only at the “Villa delle Ginestre Rehabilitation Medical Centre”, affiliated with the U.O.S.D. of Child Neuropsychiatry of the Department of Translational Medical Sciences, University of Naples Federico II. Participants belonging to the experimental group were introduced to weekly therapy based on the IAmHero tool, supplementing the other therapies already carried out as per the scheduled therapy. Each therapeutic session lasted 50 min, within which 30 min were allocated to the administration of the IAmHero tool. Specifically, the tool activities were administered for a duration of 10 min each in the respective order of Topological Categories, Infinite Runner, and Space Coding (see Section 2.3). The sessions were supervised by one of the children’s respective referring therapists (speech therapists, neuropsychomotricists, and physiotherapists). The last 20 min of the therapeutic session were designated for free play in the room. Conventional therapy consisted of weekly sessions (2 sessions per week) of speech and/or psychomotor treatment, prescribed on the basis of the patient’s clinical needs, at the Local Health Authority, according to an Individualised Rehabilitation Therapeutic Plan (provided by the Health System on an outpatient basis ex-article 26). All patients attending “Villa delle Ginestre Medical Rehabilitation Centre” received conventional therapy, regardless of subsequent participation in the study. After six months, a post-intervention phase was performed: a second evaluation was carried out at the end of the experimental session phase with the re-administration of specific scales and tests to define clinical outcomes (Conners 3, BIA, Tower of London test).

### 2.3. IAmHero

IamHero was created to bring to life an ambitious project in digital therapy aimed at innovating care processes for patients with neurodevelopmental disorders in pre-school, as well as those who are primary and secondary school age. The structure of the games was designed by Villa delle Ginestre’s multidisciplinary team in collaboration with the University of Naples Federico II (speech therapists, neuropsychomotricists, physiotherapists, neuropsychiatrists, neuropsychologists, phoniatrists, psychotherapists) in order to make the content easily accessible and understandable to the 5–12 age group. The IAmHero system is a serious game structured around activities that aim to implement cognitive-behavioural skills impaired in ADHD subjects. Among these, we included attention, planning, critical reasoning, visual perception, visuomotor skills, abstract reasoning, and language skills.

In particular, three different activities were developed:(1)Topological Categories:

In this serious game, the child is immersed, thanks to a virtual reality viewer, in a scenario chosen from a classroom, a bedroom, or a garden in order to make the setting familiar and put the child at ease. Directional commands to be carried out in relation to topological categories (e.g., “Get the ball and put it under the desk”) are depicted on the screen and played as sound. The child is given the opportunity to move in the virtual space in relation to the instructions given and to grasp various objects with the help of the joystick built into the virtual reality viewer.

In this way, visual-spatial orientation, motor coordination, planning, and selective auditory attention are stimulated in the child.

(2)Infinite Runner:

This type of activity is carried out with the help of a motion sensing tool, which allows recognition of the child’s body pattern and movements in order to place the child in a virtual scenario in which the child will have to avoid obstacles and collect rewards. This is to stimulate behavioural inhibition and the ability to control and manage impulses as the child, due in part to the gamification and rewards system, will have to dodge certain obstacles and functionally direct their motor activation.

(3)Space Coding:

This type of activity can be enjoyed either in VR or as mobile/tablet software. Puzzle-like tasks are proposed that will orient the child to reconstruct sequences or objects, either by manipulating tiles (fine motor skills) or by swiping them. This trains planning, visuospatial and constructive skills, as well as reasoning and problem-solving.

The system also contains a platform for therapy session management. This platform consists of an application that allows the therapist to monitor the progress of sessions, assess the achievement of goals, and tailor therapy accordingly. The combined use of these elements allows the patient to enjoy an innovative therapy service, both in the clinic and at home.

Regarding the instrumentation used, the type of hardware used was the Oculus Quest 2. Specifically, thanks to an integrated mirroring system (AirLink) and the use of a platform for loading games (Steam), the game was started from the therapist’s PC and projected directly into the virtual reality viewer. By doing so, the therapist was able to visualize in real-time what was happening in the viewer so that interactions could also be directed in a more hands-on and free manner.

### 2.4. Statistical Analysis

The results were analysed using IBM Statistical Package for Social Science Software (25th version) [23] for descriptive and inferential statistical analysis to assess the following outcomes: decrease in scores on the Conners 3 scale and improvements in BIA and TOL test scores.

### 2.5. Ethical Considerations

The present study was conducted after approval of the study protocol by the ethics committee of the Federico II University of Naples. Both participants and parents/legal guardians were informed about the modalities of the study as well as the aim and purpose before enrollment. Informed consent was obtained from each of the participants. Participation in the study was free and unconditional; in addition, one’s consent to participate could be withdrawn at any time. The collection and analysis of the data exhibited were conducted anonymously, respecting the privacy of each participant.

## 3. Results

### 3.1. Primary Endpoint: Decrease in Scores on Conners 3 Scales

With regard to the behavioural indicators of the Conners-3 scales, the following variables were identified:-Inattention;-Hyperactivity/impulsivity;-Learning problems;-Provocation/aggressiveness;-Relations with family.

Taking advantage of the internal reliability of the test, an initial analysis was carried out to investigate the correlation coefficient between the variables at T0 and T1; results are reported in Table 2.

From the analysis conducted, as also summarized in Table 2, the following results are of relevance:-A significant positive correlation exists between the scores of the variable “Hyperactivity/impulsivity” at T0 and T1;-A significant positive correlation between the scores related to the variable “Learning problems” at T0 and T1;-A significant positive correlation between the scores related to the variable “Provocation/aggressiveness” at T0 and T1;-A significant positive correlation between the scores related to the variable “Relationship with family members” at T0 and T1.

A descriptive analysis of the averages of these variables at T0 and T1 was also conducted (Table 3); moreover, a comparison of averages of the scores for paired samples was conducted (Table 4).

Lastly, the reliability change index (RCI) was calculated to assess the statistical reliability of the change due to the treatment (Table 5). In particular, this analysis was carried out to assess whether the individual variations highlighted were truly statistically significant or derived, for example, due to random error in measurement.

The following results emerged:

The contribution of treatment is clear considering the change in the respective mean values of the T-scores of variables of the Conners-3 scale, particularly regarding hyperactivity/impulsivity, learning-related problems, and relationships with family members. However, the RCI value is only fully reliable in a range >1.96, so the prospects certainly include the proposal to extend the trial to a larger sample.

### 3.2. Primary Endpoint: Improvement in BIA and TOL Test Scores

With regard to the cognitive indicators measured by mean scores of the BIA and TOL tests, the following variables were identified for BIA:-Sustained visual attention (identified from the number of omissions in the continuous performance subtest);-Impulsive behavior (calculated from the number of errors in the MF subtest);-Inhibitory control (identified with the response time of the MF subtest);-Selective auditory attention (represented by the auditory attention test score);-Motor inhibition (obtained from the frogs test score).

The following variables were identified for TOL:-Organisation and planning (representing the number of moves score);-Problem-solving (obtained from resolution time);-Executive functions (calculated from the main score of the test).

A correlation analysis was conducted between the BIA variables at T0 and T1 and between the TOL variables at T0 and T1. The results are reported in Table 6 and Table 7.

We highlighted the following results:-A significant positive correlation between the values of the scores for the variables “Sustained Visual Attention”, “Impulsive Behaviour”, “Inhibitory Control”, and “Auditory Selective Attention” of the BIA test at T0 and T1;-A significant positive correlation between the values of the scores for the variables “Organisation and planning”, “Problem solving”, and “Executive functions” of the TOL test at T0 and T1.

This shows that there is a relationship between the values at the initial and post-treatment time. From this initial observation, we examine the weight of the relationship between the variables at the two different times by analyzing the difference between the mean scores obtained in the tests in order to establish their significance in terms of improvement.

A descriptive analysis of the averages of these variables at T0 and T1 was also conducted (Table 8 and Table 9); moreover, a comparison of the averages of the scores for paired samples was conducted (Table 10 and Table 11).

Our statistical analysis shows a significant improvement in the areas of attentional processes, problem solving, sustained auditory attention, and task planning and organisation and, more generally, an improvement in executive functions. Last, we proceeded to calculate the RCI value for the individual variables (Table 12).

## 4. Discussion

The study, we analysed the benefits of using the serious game approach in the rehabilitation of children with ADHD between the ages of 5 and 12. From what emerged, there are benefits both on the core symptomatology of ADHD and on further neuropsychological functions assessed with standardised instruments. In particular, it is well established that general executive dysfunction is present in subjects with ADHD: the domains most affected are task monitoring, preparatory processing and response inhibition [24]. Related to this topic, in our study, we can see that the performance of the treated children improved on the TOL test, which is specifically used to assess executive functions in our sample. These findings are of great impact when one considers that impaired executive functions in individuals with ADHD may predict a worse academic outcome in later life [25]. One of the innovations of the study conducted also concerns the use of VR in the rehabilitation of individuals with ADHD. In fact, such treatments have recently been introduced in this field, and the literature shows that they are effective in the treatment of ADHD symptoms, mainly in attentional functions [19,20,21]. The results obtained from our sample agree with what is found in the literature. Furthermore, given what has been said about the need for an integrated approach, some authors have also pointed out how rehabilitation using virtual tools can be as effective as drug therapy with methylphenidate [26]. From what is present in the literature, it is mainly denoted how there is much discrepancy both in terms of the devices used and the assessment instruments; to date, therefore, this aspect remains one of the main critical issues in this field. In our study, we expose the preliminary results obtained in our sample: specifically, only the data within the group of treated children have been analyzed so far; the comparison data with respect to the control group are currently still under investigation. It is precisely this that is identified as one of the main criticality to the various studies in the literature and to which we would like to extend the evaluation in order to expand the evidence on the topic. Apart from the beneficial effects, it is also evident that patients show good compliance with this type of treatment and that, in general, there is good acceptance of these new methods [21]. Moreover, VR, in light of its flexibility, has also shown beneficial effects in the treatment of adults with ADHD [27] and, in general, in the rehabilitation of different psychiatric disorders [28].The use of new technologies is an ever-expanding field, and using these tools in the rehabilitation of childhood neuropsychiatric disorders could, in part, constitute a breakthrough. Unfortunately, to date, the use of protocols with little consensus and the paucity of validation studies available make the use of such tools still experimental. Specifically, our study also has several limitations, including a low sample size and a short trial time. Therefore, it would be useful to plan future studies with as little bias and limitations as possible.

## 5. Conclusions

Given the findings from the analysis of the data collected during the experimental phase, we can state that the virtual reality therapeutic tool “IAmHero” results in improvements in the treatment of attention and/or hyperactivity disorder. In particular, improvements can be seen from the behavioural point of view regarding levels of hyperactivity/impulsivity, improvements in learning-related issues, and improvements in relationships with family members. On the cognitive side, there is evidence of many improvements from the perspective of executive functions and, more specifically, improvements in task planning and organization, sustained auditory attention, problem-solving, and management of impulsive behaviours. There also turns out to be a statistically significant RCI in improving selective auditory attention. Given the strong impact of ADHD symptomatology and executive function deficits on the adaptive functioning of individuals with this disorder, it is necessary to provide rehabilitation as early and effectively as possible. A potential improvement in the experimental design could concern expanding the proposed set of activities to include the treatment of additional cognitive functions and to broaden the therapeutic target since, as is also evident from the literature, deficits in executive functions are common and recurrent in different types of neurodevelopmental disorders. However, several weaknesses should be considered that may affect the collected data and, consequently, should be controlled for even more effective future experimentation. In particular, it will be necessary to:-Expand the reference sample to enjoy a larger dataset and move in the direction of standardizing the instrument;-Provide, in a hypothetical subsequent trial, for the setting of an additional follow-up phase to monitor treatment progress and compare it with the data collected by the management platform;-Conduct the study over a longer time frame in order to monitor the longitudinal effects of the virtual reality intervention. Despite this, for the time being, the results obtained both from our study and from other studies in the literature appear encouraging. We, therefore, consider it useful to continue experiments in this field in order to develop a new resource in the treatment of ADHD;-Evaluate the acceptability of such tools by children and especially better define what contribution they can provide in terms of collaboration in rehabilitation settings.

## Figures and Tables

**Table 1 ijerph-20-03414-t001:** Eligibility criteria used at recruitment stage.

Diagnosis of ADHD in accordance with DSM-5 diagnostic criteria made with standardized scales with/without neuropsychiatric comorbidities (learning disorder, language disorder)
Chronological age between 5 and 12 years
Rehabilitation plan in accordance with the goals of the intervention
Native Italian-speaking children or adolescents
Written informed consent from parents or legal guardians

**Table 2 ijerph-20-03414-t002:** Correlation study between the Conners-3 scale variables at T0 and T1.

	Inattention (T1)	Hyperactivity/ Impulsivity (T1)	Learning Problems (T1)	Provocation/ Aggressiveness (T1)	Relations with Family (T1)
Inattention (T0) Sig. (2-tailed)	0.319 0.197				
Hyperactivity/Impulsivity (T0) Sig. (2-tailed)		0.618 * 0.006			
Learning Problems (T0) Sig. (2-tailed)			0.539 * 0.021		
Provocation/Aggressiveness (T0) Sig. (2-tailed)				0.630 * 0.012	
Relations with Family (T0) Sig. (2-tailed)					0.507 * 0.032

* Significant at level 0.05.

**Table 3 ijerph-20-03414-t003:** Descriptive analysis of the averages of the T-scores obtained for the Conners-3 scale variables at T0 and T1.

Paired Variables	Mean	Standard Deviation
Inattention (T0) Inattention (T1)	63.72 60.78	9.58 5.90
Hyperactivity/Impulsivity (T0) Hyperactivity/Impulsivity (T1)	65.68 59.78	8.50 3.78
Learning Problems (T0) Learning Problems (T1)	65.17 59.72	10.52 5.10
Provocation/Aggressiveness (T0) Provocation/Aggressiveness (T1)	57.17 50.72	18.04 5.03
Relations with Family (T0) Relations With Family (T1)	60.22 56.17	13.81 4.72

**Table 4 ijerph-20-03414-t004:** T-test for mean differences for paired samples of the T-scores on the Conners-3 scale (T0 − T1).

	Mean Difference	Sign. (2-Tailed)
Inattention (T0) − Inattention (T1)	2.94	0.208
Hyperactivity/Impulsivity (T0) − Hyperactivity/Impulsivity (T1)	5.90	0.030
Learning Problems (T0) − Learning Problems (T1)	5.45	0.007
Provocation/Aggressiveness (T0) − Provocation/Aggressiveness (T1)	6.45	0.000
Relations with Family (T0) − Relations with Family (T1)	4.05	0.012

**Table 5 ijerph-20-03414-t005:** Summary table of the RCI values of the scores at Conners-3 scale variables.

	RCI
Inattention	1.10
Hyperactivity/impulsivity	1.82
Learning problems	1.95
Provocation/aggressiveness	1.55
Relationships with family	1.88

**Table 6 ijerph-20-03414-t006:** Correlation study between variables at T0 and T1 of the BIA test.

	Sustained Visual Attention (T1)	Impulsive Behaviour (T1)	Inhibitory Control (T1)	Selective Auditory Attention (T1)	Motor Inhibition (T1)
Sustained visual attention (T0) Sig. (2-tailed)	0.617 * 0.006				
Impulsive behaviour (T0) Sig. (2-tailed)		0.525 * 0.039			
Inhibitory control (T0) Sig. (2-tailed)			0.513 * 0.004		
Selective auditory attention (T0) Sig. (2-tailed)				0.781 * 0.001	
Motor inhibition (T0) Sig. (2-tailed)					0.425 0.215

* significant at level 0.05.

**Table 7 ijerph-20-03414-t007:** Correlation study between variables at T0 and T1 of the TOL test.

	**Organisation and Planning (T1)**	**Problem-Solving (T1)**	**Executive Functions (T1)**
Organisation and planning (T0) Sig. (2-tailed)	0.632 * 0.013		
Problem-solving (T0) Sig. (2-tailed)		0.698 * 0.004	
Executive functions (T0) Sig. (2-tailed)			0.687 * 0.035

* significant at level 0.05.

**Table 8 ijerph-20-03414-t008:** Descriptive analysis of the averages of the scores at the variables of the BIA test at T0 and T1.

Variables	Mean	Standard Dev.
Sustained visual attention T0 Sustained visual attention T1	16.63 9.15	12.44 5.10
Impulsive behaviour (T0) Impulsive behaviour (T1)	20.25 13.61	3.14 2.11
Inhibitory control (T0) Inhibitory control (T1)	18.97 15.25	3.87 2.27
Selective auditory attention (T0) Selective auditory attention (T1)	7.21 9.00	0.65 0.36
Motor inhibition (T0) Motor inhibition (T1)	21.44 22.13	8.44 6.68

**Table 9 ijerph-20-03414-t009:** Descriptive analysis of the averages of the scores at the variables of TOL test at T0 and T1.

Variables	Mean	Standard Dev.
Organisation and planning (T0) Organisation and planning (T1)	94.05 79.52	12.76 5.47
Problem-solving (T0) Problem-solving (T1)	377.24 266.0	42.92 35.75
Executive functions (T0) Executive functions (T1)	24.45 27.79	5.65 3.66

**Table 10 ijerph-20-03414-t010:** Student’s *t*-test for difference between averages for paired samples of BIA test variables.

	Mean Differences	Sign. (2-Tailed)
Sustained visual attention (T0) − Sustained visual attention (T1)	7.48	0.001
Impulsive behaviour (T0) − Impulsive behaviour (T1)	6.64	0.045
Inhibitory control (T0) − Inhibitory control (T1)	3.72	0.513
Selective auditory attention (T0) − Selective auditory attention (T1)	−1.79	0.001
Motor inhibition (T0) − Motor inhibition (T1)	−0.69	0.853

**Table 11 ijerph-20-03414-t011:** Student’s *t*-test for difference between averages scores for paired samples of the TOL test variables.

	Mean Differences	Sign. (2-Tailed)
Organisation and planning (T0 − T1)	14.53	0.013
Problem solving (T0 − T1)	11.24	0.004
Executive functions (T0 − T1)	−3.34	0.035

**Table 12 ijerph-20-03414-t012:** Summary table of RCI values of BIA and TOL test variables.

	RCI
Sustained visual attention	1.89
Impulsive behavior	1.82
Inhibitory control	1.11
Selective auditory attention	1.99 *
Motor inhibition	0.77
Planning and organisation	1.79
Problem solving	1.94
Executive functions	1.68

* significant for the range > 1.96.

## Data Availability

The data used for the study are available upon reasonable request.

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
