# Peer review of "IAmHero: Preliminary Findings of an Experimental Study to Evaluate the Statistical Significance of an Intervention for ADHD Conducted through the Use of Serious Games in Virtual Reality"

_ijerph, 2023, doi:10.3390/ijerph20043414_

Round 1

Reviewer 1 Report

The study is well-designed and the results are interesting.

The only concern is kids with ADHD usually fall in the spectrum and might have some aversion towards flickering lights and having the VR headset on their heads (which is another thing, couldn't find any information on whether it was a VR headset or was it through some other way).

Also, an evaluation of how the children felt about the games itself is also missing, like a usability evaluation or such.

Reviewer 2 Report

An experimental study was conducted to assess the statistical significance of an intervention for ADHD through the employment of Serious Games in Virtual Reality. The IAmHero system is a serious game structured around activities that aim to implement cognitive-behavioral skills impaired in ADHD subjects. Participants were randomly divided into two groups, one experimental and one control, who were given virtual reality therapies with the IAmHero system alternating with traditional therapy and traditional therapies. The use of the IAmHero program in virtual reality on a group of participants with ADHD aged 5 to 12 revealed improvements in both ADHD symptoms (especially in the hyperactivity/impulsivity domain) and executive functions.

The topic is interesting. Indeed, there is a growing interest regarding the role of serious games and virtual reality in the intervention of people with ADHD. The quality of the study is good enough with valuable results.

I would like to make some suggestions:

·        I suggest mentioning in the title type filed, the type of the paper. In addition, in the title, it would be better to clarify the accurate type of this experimental study.

·        In the introduction, it is important to highlight the objectives of this study, the hypothesis as well as the reason that explains the importance of the study and its added value.

·        In the introduction, it is important to discuss the existing literature on this intervention and better clarify the existing gaps.

·        In the methodology section, it would be useful to provide additional information. For instance, about the duration of the sessions, the staff which was responsible for the training.

·        In keywords, please remove duplicates.

·        It is suggested to use the journal’s template for the tables.

·        It would be useful to add information about the type of virtual reality equipment.

·        Please better organize text, with paragraphs and other formatting changes according to the journal’s template and guidelines. For eligibility criteria, it would be better to use a table. Several paragraphs are too short without inappropriate punctuation and telegraphic statements.

·        Results could be described in more detail within paragraphs.

·        In the discussion session, it would be better to omit introductory information (i.e at the beginning of the section) and focus, for instance,  on more critical comments considering other relevant studies.

·        In the text, reference numbers should be placed in square brackets [ ], and placed before the punctuation; for example [1], [1–3], or [1,3]. For embedded citations in the text with pagination, use both parentheses and brackets to indicate the reference number and page numbers; for example [5] (p. 10). or [6] (pp. 101–105).

Reviewer 3 Report

I find the manuscript well written and it contains interesting results on novel rehabilitation methods via VR for children having ADHD. The data consist of 60 children aged 5-12 years from two “medical centers”.  The main findings  is that the use of “IAMHero” therapeutic VR tool shows improvements in the treatment of ADHD. The analysis is carried out in separately for variables showing improvements in behavior and improvements in cognitive functions. The improvements in a behavioral point of view in levels of hyperactivity/impulsivity, improvements in learning- related issues as well as family relations. Improvements in cognitive functions were observed for task planning, auditory attention, problem solving and management of impulsive behaviors. 

  • General concept comments

As the authors point out, some weaknesses with the study are the relative small sample size and the follow-up period.  Nevertheless I find this study well conducted anf following ethical guidelines.  My reflection concerns the patients included in this study. Did the authors take possible use of medication into account when recruiting the children having ADHD? Is it possible that a change in drugs could give the same positive results as shown for children receiving “IAMHero”?

I also think that the manus would improve of including more details about the “IAMHero” therapeutic tool. When and where was this tool created and by whom? Is it the same tool used for children who were 5 years as those who were 12 years?

  • Specific comments 

I think that that the results in the manus would be easier to follow if the “variable names” were replaced by text in all tables, e.g. APRP_CONN3 is replaced to Learning problems. The statistical analysis section may be improved by adding a description of the RCI analysis.

How does previous research on the topic relate to your specific findings? Do they confirm your findings? In the discussion the authors state that several studies found that AI is effective, ref 23,24,25.
